# Impact of Bacterial Infections on COVID-19 Patients: Is Timing Important?

**DOI:** 10.3390/antibiotics12020379

**Published:** 2023-02-12

**Authors:** Christos Michailides, Themistoklis Paraskevas, Iosif Karalis, Ioanna Koniari, Charalampos Pierrakos, Vasilios Karamouzos, Markos Marangos, Dimitrios Velissaris

**Affiliations:** 1Department of Internal Medicine, University Hospital of Patras, 26223 Patras, Greece; 2Cardiology Department, Liverpool Heart and Chest Hospital, Liverpool L14 3PE, UK; 3Department of Intensive Care, Brugmann University Hospital, Université Libre de Bruxelles, 1050 Brussels, Belgium; 4Intensive Care Unit, University Hospital of Patras, 26223 Patras, Greece

**Keywords:** COVID-19, co-infections, superinfections, infection timing

## Abstract

Background: Along with important factors that worsen the clinical outcome of COVID-19, it has been described that bacterial infections among patients positive for a SARS-CoV-2 infection can play a dramatic role in the disease process. Co-infections or community-acquired infections are recognized within the first 48 h after the admission of patients. Superinfections occur at least 48 h after admission and are considered to contribute to a worse prognosis. Microbiologic parameters differentiate infections that happen after the fifth day of hospitalization from those appearing earlier. Specifically, after the fifth day, the detection of resistant bacteria increases and difficult microorganisms emerge. Objectives: The aim of the study was to evaluate the impact of bacterial infections in patients with COVID-19 on the length of the hospital stay and mortality. Methods: A total of 177 patients hospitalized due to COVID-19 pneumonia were consecutively sampled during the third and fourth wave of the pandemic at a University Hospital in Greece. A confirmed bacterial infection was defined as positive blood, urinary, bronchoalveolar lavage (BAL) or any other infected body fluid. Patients with confirmed infections were further divided into subgroups according to the time from admission to the positive culture result. Results: When comparing the groups of patients, those with a confirmed infection had increased odds of death (odds ratio: 3.634; CI 95%: 1.795–7.358; *p* < 0.001) and a longer length of hospital stay (median 13 vs. 7 days). A late onset of infection was the most common finding in our cohort and was an independent risk factor for in-hospital death. Mortality and the length of hospital stay significantly differed between the subgroups. Conclusion: In this case series, microbial infections were an independent risk factor for a worse outcome among patients with COVID-19. Further, a correlation between the onset of infection and a negative outcome in terms of non-infected, community-acquired, early hospital-acquired and late hospital-acquired infections was identified. Late hospital-acquired infections increased the mortality of COVID-19 patients whilst superinfections were responsible for an extended length of hospital stay.

## 1. Introduction

COVID-19 is a widespread infection that caused a pandemic in 2019, leading to high mortality rates [1]. Several risk factors are considered to aggravate the final outcome such as chronic obstructive pulmonary disease, chronic kidney disease, cardiovascular disease, diabetes mellitus, obesity and age [2]. Patients with COVID-19 frequently develop acute respiratory distress syndrome (ARDS) and an abnormal response to the infection, leading to a cytokine storm. This rouses the urgency to use corticosteroids and other immunosuppressive or immunomodulating drugs in hospitalized patients [3,4,5]. Infection from SARS-CoV-2 can lead to severe immunosuppression [6]. This actuated researchers to examine the effect of microbial infections in those patients, deducing that they lead to increased mortality and an increased length of hospital stay (LOHS) [7,8,9]. A community-acquired (CA) infection or co-infection is defined as the occurrence of a positive culture with a clinically significant microorganism within the first 48 h of hospital admission. A hospital-acquired infection (HAI) is defined as a positive culture after the first 48 h. A few studies compared the impact of CA and HA infections with mortality and the LOHS. The magnitude of this separation is that the microbiology is different between those groups, with more multi-drug-resistant (MDR) microorganisms isolated from hospital-acquired (HA) infections. This leads to difficulties to treat them and, subsequently, worse clinical manifestations and outcomes such as septic shock and increased mortality, respectively [10,11]. Alterations to the microbiologic profile of infections have also been detailed among patients with nosocomial pneumonia, comparing infections before (early hospital-acquired (EHA)) and after day 5 (late hospital-acquired (LHA)) at the point of diagnosis of non-COVID-19 [12] and COVID-19 patients [13,14]. Our data, which described the changes in the outcomes of patients due to alterations to the microbiology throughout hospitalization, could lead physicians to prescribe apt antibiotics to COVID-19 patients according to the day of a new possible infection initiation. The design of this study—to examine how timing affected the outcomes of those patients—was based on the urge to guide the prescription of antibiotics to COVID-19 patients and improve clinical practice.

## 2. Methods

### 2.1. Setting

This was a retrospective observational cohort study that referred to the third and fourth wave of the COVID-19 pandemic in Greece. All patients who were hospitalized for COVID-19 pneumonia between November 2021 and March 2022 in the Department of Internal Medicine at the University Hospital of Patras, Greece, were assessed for eligibility. The data were extracted from the electronic hospital files of the patients. The Ethics Committee of the University Hospital approved the study protocol.

### 2.2. Participants

All consecutive patients were assessed to meet the following inclusion and exclusion criteria. The inclusion criteria were an age > 18, a positive real-time polymerase chain reaction (RT-PCR) for SARS-CoV-2 and clinical or radiological evidence of pneumonia. The exclusion criteria were an age < 18, pregnancy, a preference of the patient for exclusion from epidemiological studies and an expected length of stay < 24 h. Patients were followed-up until they were discharged from the hospital.

### 2.3. Treatment 

Patients were treated for COVID-19 according to national protocols, receiving remdesivir, dexamethasone, low-molecular-weight heparin (LMWH) and immunomodulatory (tocilizumab/baricitinib) medication as per the need of each case, except for those with a contraindication.

### 2.4. Definitions

A community-acquired (CA) infection or co-infection is an infection detected within the first 48 h of hospital admission. A superinfection or hospital-acquired infection (HAI) refers to all infections detected after the first 48 h. Further, early hospital-acquired (EHA) infections are the ones detected between the second and fifth day of hospitalization, whilst late hospital-acquired (LHA) infections are those detected after the fifth day. A confirmed infection was considered to be a positive blood, urinary, BAL, sputum or pleural fluid culture or a positive stool for the *Clostridium difficile* antigen or toxin. Non-infected patients were those with no clinical signs and symptoms of infections of any origin or any laboratory tests, including cultures that indicated an infection as well as COVID-19. Vaccinated patients were those who had received the required number of vaccine doses that were mandatory according to the National Committee of Vaccination during the period of their hospitalization. Most of the vaccinated patients had received three doses at that time.

### 2.5. Data Extraction

For each patient, the following data were collected: demographics; vital signs in the emergency department (temperature, blood pressure, heart rate, oxygen saturation, respiratory rate and PO_2_/FiO_2_); arterial blood gases in the emergency department (PaO_2_, PaCO_2_, pH, HCO^3−^ and lactate); and routine laboratory data on admission. 

### 2.6. Statistical Methods

All data were presented as the mean ± standard deviation for the continuous variables with a normal distribution, as the median and interquartile range for the continuous variables with a non-normal distribution or as the frequency and percentage for the categorical variables. The Mann–Whitney U- and Kruskal–Wallis H-test were used to compare the non-parametric continuous data. The Student’s *t*-test was used to compare the data with a normal distribution and the chi-2 test was used for the categorical data. IBM SPSS Statistics software (version 27) and Microsoft Excel were used for the data analysis and graph design.

## 3. Results

A total of 177 patients were included in the study. The mean age of the enrolled patients was 67.4 years (SD: 16.5); 80 of the 177 (45.2%) were women. PO2 referred to the ambient air measurement. A total of 53 (29.9%) patients died during hospitalization. The demographics upon the admission of our cohort are presented in Table 1.

A total of 47 patients had at least 1 documented co-infection or superinfection on top of COVID-19; 24 of whom died (51.1%). Patients with such infections had increased odds of death (odds ratio (OR): 3.634; CI 95%: 1.795–7.358; *p* < 0.001) and a longer length of hospital stay (median 13 vs. 7 days).

Additionally, we categorized the patients according to the presence and timing of the infection into the following groups: (1) no infection; (2) community (<48 h); (3) early infection (48 h–5 d); and (4) late (>5 d). The mortality significantly differed between the subgroups (Table 2).

In the univariate logistic regression using no infection as the reference category, only the presence of a late infection was associated with a significant increase in mortality (odds ratio 5.805; 95% CI: 2.541–13.262; *p* < 0.001). A late infection remained an independent risk factor after an adjustment for the 4C score on admission, which incorporated the demographic, clinical and laboratory values (*p* < 0.001; adjusted OR: 6.435 [2.434–17.015]).

The total LOHS significantly differed between the subgroups (*p* < 0.001). The median LOHS was similar in the no infection/community infection groups and the early/late infection groups, with the latter two having a longer LOHS (Figure 1).

The epidemiological data regarding those groups were also extracted. Among the patients with a documented infection on top of COVID-19 within the first two days of hospitalization, three cultures were positive for *E. coli*, two for *Coagulase-Negative Staphylococci (CNS)* and one each for *Streptococcus viridians*, *Enterococcus faecium*, *Acinetobacter baumanii* and *Klebsiella pneumoniae*. *CNS* were the most frequent bacteria among the second group of patients and the occurrence of *Pseudomonas aeruginosa* and *Providencia stuartii* was observed. In the late infection group, *Klebsiella* spp., *Acinetobacter baumanii* and *Candida* spp. dominated (Figure 2, Figure 3 and Figure 4).

## 4. Discussion

SARS-CoV-2 causes an infection that dysregulates the immune system and triggers various systematic manifestations such as ARDS and sepsis, but also causes histopathologic changes such as lung cellular damage, fibrosis and cytokine overproduction [3,6]. The abnormal response to pathogens and the consequent hyper-inflammation urges the use of anti-inflammatory drugs and blocks inflammation mediators such as cortizole, IL-1 receptor blockers and IL-6 receptor blockers [3,4,5]. Immune dysregulation mostly pertains to an adaptive immunity dysfunction, consequently (and primarily) causing lymphopenia [6]. The aforementioned pathological alterations, in combination with common clinical practices such as central venous catheterization, are the fundamentals for co-infections and superinfections. The combination of cortizole plus anti-IL-6 or anti-IL-1 agents has been proven to be significantly predisposed to blood stream infections (BSI) in COVID-19 patients [15]. Several studies have described that the microbiology differs between a CAI and a HAI. Infections detected within the first 48 h from admission are most commonly caused by *E. coli, Klebsiella, Haemophilus influenza*, *Staphylococcus aureus*, *Streptococcus pneumoniae* and *Staphylococcus epidermidis* species [7,10,16,17]. Hospital-acquired infection pathogens are stated to differ based on the timing of positive cultures [8,16]. In general, a higher percentage of MDR organisms has been observed, along with an increased number of *Coagulase-Negative Streptococci (CNS)*, *Methicillin-Resistant Staphylococci Aureus (MRSA), Vancomycin-Resistant Enterococci (VRE)*, *Acinetobacter*, *Pseudomonas* and *Stenotrophomonas species* [7,10,11,16,18].

Actuated by the current literature, we retrospectively examined how co-infections and superinfections affected our COVID-19 patients in terms of mortality and the length of hospital stay. A total of 47 of the 177 patients were infected (26.6%). We found that 51.1% of the infected patients died vs. 22.3% of the non-infected (OR = 3.634), which was statistically significant. We extrapolated that the infections were an independent risk factor of death in COVID-19 patients. An English observational study found a similar incidence of co-infections/superinfections among COVID-19 patients [16]. A systematic review and meta-analysis by Musuuza et al. estimated that the death OR was equal to 3.31; this included studies on COVID-19 patients complicated with microbial co-infections or superinfections between 2019 and 2021 [7]. Our data confirmed that this estimation applied to our cohort of patients. A mini-review by Feher et al. demonstrated a similar mortality among infected patients (53.1%) despite an increased mortality among non-infected patients compared with our findings [17]. A meta-analysis by Langford et al. [19] estimated a lower incidence of co-infections, reporting 281 infections out of 3338 tested (8.41%). In our study, the median LOHS of non-infected patients was 7 days vs. 13 days for the infected patients. Our data agreed with the literature on the LOHS of non-infected patients, but presented a lower hospital LOHS for the infected patients. This may have been due to an early IV antibiotic administration initiation on day 1 in the emergency room to all patients admitted to our hospital with signs of an infection before positive COVID-19 PCR results were available [8,16,17] and a continuation until negative cultures were reported to cover the contingency of bacterial infections. For that reason, fewer additional days of IV antibiotics may have been needed for the cases with positive cultures. Increased mortality may occur due to infection complications such as septic shock [11] and renal dysfunction [8]. An increased LOHS may be associated with the aforementioned complications, but may also be related to an increased incidence of intubation [8] and the need for a longer intravenous (IV) antibiotic administration.

We further categorized the patients into four separate groups based on the timing of detecting the infections. The differences between CA infections and HA infections have already been described in terms of a worse clinical outcome along with a different microbiology [10,11]. The need for a further separation of HA infections has been considered in previous literature about HAP as well as the current literature on COVID-19 and infections. Second-to-fifth day-acquired HAP has been demonstrated to be caused by different pathogens and to have a worse clinical outcome compared with HAP detected after the fifth day [12]. Among COVID-19 patients, the microbiology differs after the fifth day of hospitalization. Zhu et al. described an increased incidence of Cryptococcosis [13] and Hughes et al. described an increased incidence of *Enterococcus* spp., *Candida* spp., *Pseudomonas* spp., *Klebsiella* spp., *Enterobacter* spp., *Providencia* spp. and *Stenotrophomonas maltophilia* [14]. In our cohort, the non-infected group consisted of 130 patients. CA infections were observed in 9 patients, EHA in 6 patients and LHA in 32 patients of our cohort. These data were in line with other evidence that HA infections in COVID-19 patients have a median onset 6 days after admission [8].

Our data confirmed our hypothesis that LHA infections were those with a major clinical significance, as long as we proved that there was a statistical significance in mortality only between LHA infections compared with the non-infected group as a reference category (OR = 5.805). To amplify our findings, we adjusted them for the 4C score (which is built on demographic, clinical and laboratory values) used for a COVID-19 disease prognosis [20]. An adjusted OR = 6.435 proved that LHA was an independent risk factor for mortality among COVID-19 patients. The non-infected group and CA infection group had equal mortality percentage values (22.3% and 22.2%, respectively), which may have been due to susceptible pathogens in the CA infection group that were successfully treated with empiric broad spectrum antibiotics from day 1 in the emergency room. A mortality rate of 33.3% in the EHA infection group demonstrated a trend of mortality to proportionally accrete to the in-hospital days. The LOHS was similar in the non-infected and CA infection groups (median 7 days and 6 days, respectively) and also in the EHA and LHA groups (median 14.5 days and 14 days, respectively). On the contrary, it significantly differed between the non-infected and CA infection groups compared with the EHA and LHA groups. That finding was in line with the current literature, which describes that superinfections increase the LOHS compared with no infections [8]. Concurrently, our data imbue the literature with evidence that only superinfections, and not co-infections, can extend the LOHS. A possible reason could be the increased incidence of MDR organisms reported among those patients [10,11,18], which require more days of targeted IV antibiotic treatments. The epidemiological data from our study agreed with the current literature in that there are important differences in the microbiology throughout hospitalization [7,10,14,16]. Sensitive isolates occur during the first days of hospitalization, mostly *E. coli* and *CNS*. Early hospital infections are important because of the appearance of *Pseudomonas aeruginosa* and CNS, most frequently *Methicillin-Resistant Staphylococci* [21,22]. The dominance of MDR and XDR strains after the fifth day of hospitalization and the increased incidence of *Klebsiella pneumoniae, Acinetobacter baumanii* and *Candida* spp. are the key factors that worsen the outcomes of patients due to increased resistance levels [23,24,25,26,27]. In general, alterations to the microbiology throughout the hospital stay of a COVID-19 patient, especially between admission, 48 and 120 h later, can set a cornerstone for the prediction of infections and proper antibiotic treatments [10,11,13,14,18].

## 5. Limitations

The laboratory department of our hospital could not support viral panels, film-array and regular PCR tests for fungi and virus identifications. This could have resulted in an underestimation of the prevalence of infections. Co-colonization may also have been underestimated due to the acceptance of a single positive culture as an infection in cases where there was not a second culture. The sample size could not lead to safe deductions on how EHA and CA infections affected the clinical outcome. In this study, we analyzed how timing affected the LOHS and mortality in COVID-19 patients. Microbiological epidemiology and type of infection analyses are the subject of further investigations and were not the objective of the current study; however, we considered this to be a limitation of the data provided herein. 

## 6. Conclusions

Our data confirmed that microbial infections were an independent risk factor for a worse outcome among patients with COVID-19 infections. Further, we displayed a correlation between the infection timing and a negative outcome in terms of non-infected, community-acquired, early hospital-acquired and late hospital-acquired infections. From our data, we ascertained that LHA infections could increase the mortality of COVID-19 patients and superinfections (in general) were responsible for an extended LOHS. Additional research is required to confirm our data. Randomized control trials with a larger sample size could provide strong evidence regarding the prognosis of COVID-19 patients complicated with co-infections or superinfections.

## Figures and Tables

**Figure 1 antibiotics-12-00379-f001:**
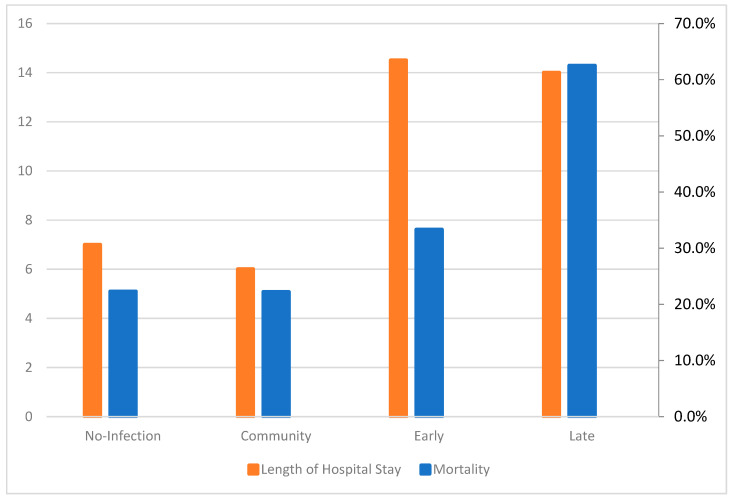
Differences in Length of Hospital Stay and Mortality between groups.

**Figure 2 antibiotics-12-00379-f002:**
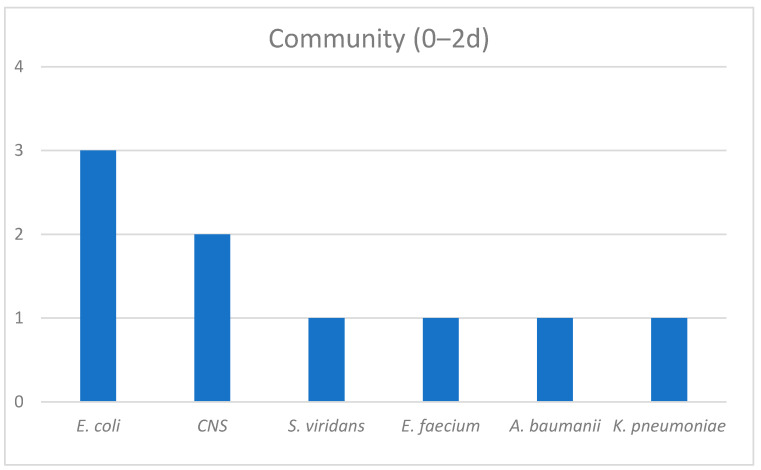
CA infections group Epidemiology.

**Figure 3 antibiotics-12-00379-f003:**
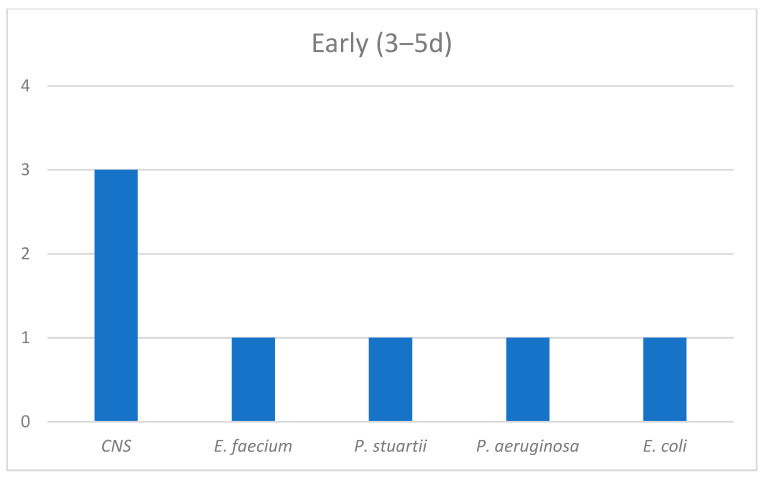
EHA infections group epidemiology.

**Figure 4 antibiotics-12-00379-f004:**
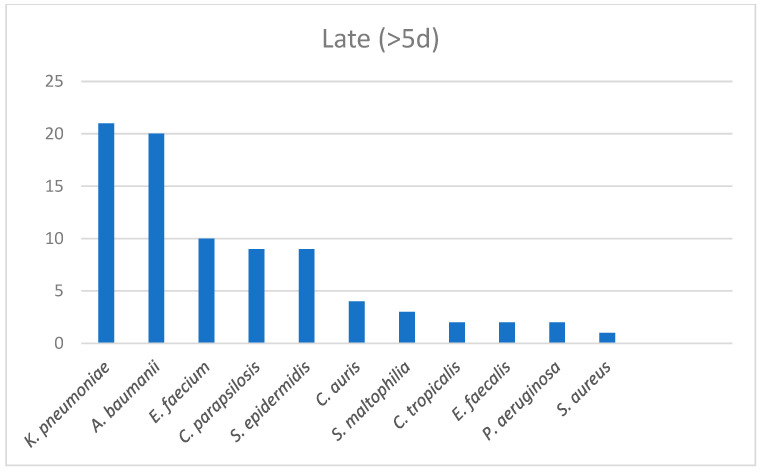
LHA group infections epidemiology.

**Table 1 antibiotics-12-00379-t001:** Demographic, clinical and laboratory characteristics of the cohort.

	Survivors (*n* = 124)	Non-Survivors (*n* = 53)	*p*-Value
Age (Years)	64.7 ± 32.2	75.2 ± 27.24	<0.001
Male (%)	68 (54.8)	29 (54.7)	0.95
Vaccinated (%)	50 (40)	13 (24.5)	0.035
4C Mortality Score	7.87 ± 8.4	13.1 ± 7.36	<0.001
PO2 (mmHg)	74.1 (21.8)	66.5 (45.5)	0.021
Lactate (mmol/L)	1.0 (0.6)	1.3 (1.08)	<0.001
WBC (K/μL)	6.05 (3.86)	8.22 (6.34)	0.005
Neutrophils (K/μL)	4.7 (4.2)	6.95 (6.55)	0.004
Lymphocytes(K/μL)	0.99 (0.75)	0.66 (0.48)	<0.001
Fibrinogen (mg/dL)	543 (183.75)	544 (178.5)	0.96
D-Dimers (μg/dL)	0.835 (1.14)	1.08 (1.41)	0.024
CRP (mg/dL)	4.15 (8.09)	9.36 (12.08)	<0.001
HS-TnI (pg/mL)	6.05 (12.35)	22.85 (37.13)	<0.001
Ferritin (ng/mL)	386 (818.23)	822 (1382.5)	<0.001

**Table 2 antibiotics-12-00379-t002:** Outcomes of patients according to their infection timing-related group.

	Mortality %	Median Length of Hospital Stay (IQR)
No Infection (*n* = 130)	22.3%	7 (6.25)
Community (*n* = 9)	22.2%	6 (9)
Early Infection (*n* = 6)	33.3%	14.5 (14.5)
Late Infection (*n* = 32)	62.5%	14 (21.5)

## Data Availability

Data are available from the corresponding author upon reasonable request.

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
