# Peer review of "Impact of Bacterial Infections on COVID-19 Patients: Is Timing Important?"

_antibiotics, 2023, doi:10.3390/antibiotics12020379_

Round 1

Reviewer 1 Report

I consider this study of importance for hospitalists and public health professionals, as these findings add to current literature. This study highlights the current reality of co-infections and superinfections in patients hospitalized with COVID19, as well as worse patient outcomes, which should be taken into account when applying infection prevention and antimicrobial stewardship practices.

In order to improve quality of presentation and help reveal the significance of study findings, I have some comments.

Please write all bacterial names in italics.

Minor improvements in language quality are needed throughout the text.

Methods

Section 2.1

Can the authors provide Ethics Committee approval if available?

Section 2.2

The Please add that included participants were patients hospitalized due to COVID19 pneumonia.

Please clarify sampling: abstract mentions this as a convenience sample, in this section patient recruitment is mentioned as consecutive.

Section 2.4

Please clarify how vaccinated patients were defined (number of doses, types of vaccines).

Please also define no-infection.

Results

Table 1: are these patient characteristics upon admission? Is PO2 on ambient air?

Can the authors provide additional information on types of infections and/or pathogens detected per infection category? Could these findings have affected patient outcomes? If this information isn’t available, then it should be mentioned as a serious limitation in study methodology and conclusions.

Table 3: this is a figure. Due to nature of data presented, please consider presenting it in bars.

Discussion

Lines 171-175: repetition.

Lines 170-209: please consider fragmenting this paragraph.

Lines 201-205: please consider rewording to improve language quality.

Additional limitations include the retrospective nature of the study and (if confirmed) the convenience sampling of patients.

Conclusion

Please consider saying “worse” outcomes instead of “bad” outcomes.

Reviewer 2 Report

-          Which microbial agents detected in infected patients? What types of infections have been detected? These data must be mentioned and discussed.

Round 2

Reviewer 2 Report

Despite the explanations of the authors, I still think that the lack of microbiological data is an important deficiency.

Author Response

Dear Reviewer, 

Considering your commentaries, we have added some epidemiological data regarding on our cohort's microbiology. We hope that you will approve our changes.

Best regards, 

Michailides Christos, MD, Corresponding Author
